



# Measurement report: Indirect evidence for the controlling influence of acidity on the speciation of iodine in Atlantic aerosols

Alex R. Baker[1], Chan Yodle[1,2]

[1]Centre for Ocean and Atmospheric Sciences, School of Environmental Sciences, University of East Anglia, Norwich NR4 7TJ, U.K.
[2]now at Department of Environmental Science, Faculty of Science and Technology, Chiang Mai Rajabhat University, Chiang Mai, Thailand 50300

*Correspondence to*: Alex R. Baker (alex.baker@uea.ac.uk)

**Abstract.** The speciation of iodine and major ion composition were determined in size-fractionated aerosols collected during the AMT21 cruise between Avonmouth, UK and Punta Arenas, Chile in September - November 2011. The proportions of iodine species (iodide, iodate and soluble organic iodine (SOI)) varied markedly between size fractions and with the extent to which the samples were influenced by pollutants. In general, fine mode aerosols (< 1 µm) contained higher proportions of both iodide and SOI, while iodate was the dominant component of coarse (< 1 µm) aerosols. The highest proportions of iodate were observed in aerosols that contained (alkaline) unpolluted seaspray or mineral dust. Fine mode samples with high concentrations of acidic species (e.g. non-seasalt sulfate) contained very little iodate and elevated proportions of iodide and SOI. These results are in agreement with modelling studies that indicate that iodate can be reduced under acidic conditions and that the resulting hypoiodous acid (HOI) can react with organic matter to produce SOI and iodide. Further work that investigates the link between iodine speciation and aerosol pH directly, as well as studies on the formation and decay of organo-iodine compounds under aerosol conditions, will be necessary before the importance of this chemistry in regulating aerosol iodine speciation can be confirmed.



# 1 Introduction

Iodine (I) plays a significant role in the destruction of ozone ($O_3$) in the atmosphere (Davis et al., 1996; Saiz-Lopez et al., 2012), being responsible for ~30% of $O_3$ loss in the marine boundary layer (MBL) (Prados-Roman et al., 2015). Oceanic

emission, principally of volatile $I_2$ and HOI formed through the reaction of $O_3$ with iodide ($I^-$) at the sea surface (Carpenter et al., 2013), is the main source of iodine to the atmosphere (Saiz-Lopez et al., 2012). Intensive iodine emissions, especially in coastal locations, can lead to bursts of new aerosol particle formation and, in some cases, the formation of cloud condensation nuclei (O'Dowd et al., 2002; Whitehead et al., 2010).

Uptake into the aerosol phase removes iodine species from gas phase ozone destruction cycles and influences the

atmospheric lifetime of iodine. However there is a complex aqueous phase chemical cycling of iodine in aerosol and some of the species involved have the potential to recycle back into the gas phase (Vogt et al., 1999; Pechtl et al., 2007). A complete knowledge of the speciation of iodine in the aerosol phase, and the factors that control this, is therefore required in order to understand the atmospheric chemistry of iodine and its impact on ozone.

Studies of aerosol I speciation over the ocean have reported the presence of iodate ($IO_3^-$) and $I^-$, as well as a very poorly

characterised fraction referred to as soluble organic iodine (SOI) (e.g. Wimschneider and Heumann, 1995; Baker, 2004, 2005; Lai et al., 2008; Lai et al., 2011; Yodle and Baker, 2019). The organic fraction is typically determined as the difference between measurements of total soluble iodine (TSI) and the sum of $I^-$ and $IO_3^-$, although recent work has started to identify its individual components (Yu et al., 2019).

A variety of methods have been used in the determination of aerosol iodine speciation, including, in a number of cases,

exposure to varying periods of ultrasonic agitation during aqueous extraction. However, previous studies have concluded that ultrasonic agitation leads to changes in aerosol iodine speciation (Baker et al., 2000; Xu et al., 2010; Yodle and Baker, 2019). Yu et al. (2019) recently demonstrated that addition of iodide and hydrogen peroxide ($H_2O_2$) to a low-iodine aerosol sample generated a large number of organic iodine species. Since acoustic cavitation during ultrasonication generates $H_2O_2$ and other reactive oxygen species (Kanthale et al., 2008), the result of Yu et al. is further evidence that the use of ultrasonic

agitation is likely to alter both inorganic and organic speciation of iodine prior to its determination. This raises the possibility that much of the published literature on I speciation in aerosols over the oceans is potentially unreliable. This situation may have contributed to the current lack of coherent understanding of the influences and controls on aerosol I speciation and its impacts on ozone chemistry in the MBL (Saiz-Lopez et al., 2012).

Using results from a one-dimensional MBL model (MISTRA), Pechtl et al. (2007) suggested a significant role for acidity in

controlling iodine speciation in sulfate (fine) and seasalt (coarse) aerosols. Their chemical mechanism incorporated reactions capable of reducing $IO_3^-$ under acidic conditions, as well as generating SOI and $I^-$ from the reaction of HOI (produced from the reduction of $IO_3^-$) with organic matter. Models that do not include these characteristics have been unable to reproduce the observed inorganic and organic speciation of iodine in marine aerosols (Vogt et al., 1999; McFiggans et al., 2000; Pechtl et al., 2007).



In this work aerosol I speciation was examined in samples collected over the Atlantic Ocean, using the optimised sampling and extraction methods of Yodle and Baker (2019) which avoid potential artefacts caused by ultrasonication. The major ion and soluble metal chemistry of the samples was used to gain insights into the controls on I speciation, with particular focus on the potential impacts of aerosol acidity.

## 2. Materials and Methods

### 2.1 Aerosol Sampling


Aerosol samples were collected during cruise AMT21 of the Atlantic Meridional Transect programme in 2011. RRS *Discovery* sailed from Avonmouth, UK on 29th September and reached Punta Arenas, Chile on 14th November. During the cruise, one high-volume collector (Tisch) was used to sample for iodine speciation and major ion (MI) chemistry. A corresponding set of samples for trace metal (TM) analysis was acquired simultaneously using a second collector. The collectors operated at flow rates of ~ 1 $m^3$ $min^{-1}$ and samples were changed approximately every 24 hours, or every ~ 48 hours at latitudes south of 25°S. The operation of both collectors was controlled by an automated wind sector controller, which interrupted sampling if there was a risk of the samples being contaminated by emissions from the ship's stack. Air volumes sampled varied from 576 – 2684 $m^3$ (median 1223 $m^3$) for the iodine / MI samples. Both collectors were equipped with Sierra-type cascade impactors allowing most samples to be analysed in two size fractions: the fine (< 1 µm) and coarse (> 1 µm) modes. In two cases (samples 15 and 30), the iodine / MI samples were fractionated into 7 size classes (aerodynamic cut-off boundaries 7.8, 3.3, 1.6, 1.1, 0.61, 0.36 µm) in order to examine size distribution in more detail (Baker et al., 2020). The cruise track and sample locations are shown in Fig. 1.





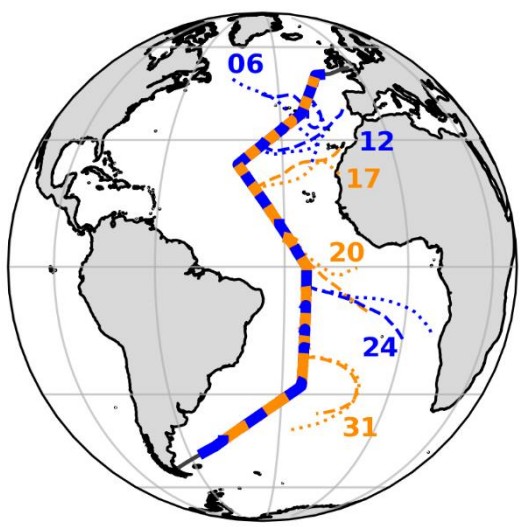

**Figure 1. AMT21 cruise track, with aerosol collection periods for each sample shown as alternating blue and orange bars. 5-day**
**airmass back trajectories for heights of 10 m (dashed) and 1000 m (dotted) above the ship's position are shown for samples 06, 12, 17, 20, 24 and 31, representing the RNA, EUR, SAH, SAF, SAB and RSA airmass types respectively (airmass codes are described in the text).**

Iodine / MI samples were collected on glass fibre (GF) substrates, while Whatman 41 substrates were used for TM samples.
Prior to use, GF substrates were washed in two ultra-high purity (UHP; 18.2 MΩ cm) water baths, dried under a laminar flow hood, wrapped in aluminium foil and ashed at 450°C for ~ 4 hours. Whatman 41 substrates were washed sequentially in 0.5 M HCl and 0.1 M $HNO_3$, dried and transferred to zip-lock plastic bags before use.

## 2.2 Extraction and Analysis of Soluble Components

For MIs and iodine species, aqueous extraction into UHP water was done using rotary mechanical agitation for 30 minutes,
followed by filtration (0.2 μm minisart, Sartorius). Analysis was by ion chromatography (IC) for MIs, inductively coupled plasma – mass spectrometry (ICP-MS) for TSI and by IC-ICP-MS for iodide ($I^-$) and iodate ($IO_3^-$). Full details of these methods can be found in Yodle and Baker (2019) and blanks and detection limits for iodine species are shown in Table 1.





**Table 1. Blanks and detection limits for aerosol iodine species for AMT21. Values are given for fine (< 1 μm) and coarse (> 1 μm) aerosol size fractions. Detection limits are presented for an air volume of 1400 m³, representative of the ~24 hour sample collection used for most of the samples.**

| | $I^-$ | | $IO_3^-$ | | TSI | |
|---|---|---|---|---|---|---|
| | < 1 μm | > 1 μm | < 1 μm | > 1 μm | < 1 μm | > 1 μm |
| **Blank (nmol/filter)** | < 0.9 | < 0.2 | < 0.3 | < 0.2 | 0.8 | 0.2 |
| **Detection Limit (pmol/m³)** | 0.7 | 0.2 | 0.2 | 0.1 | 0.5 | 0.2 |

Methods for soluble TM analysis for these samples have been reported in Baker and Jickells (2017). Briefly, these were
extraction in ~ 1 M ammonium acetate solution followed by analysis by inductively coupled plasma – optical emission spectroscopy (ICP-OES) for soluble TMs.

### 2.3 Atmospheric Concentrations

Measured aqueous phase concentrations were converted into atmospheric concentrations, taking account of the volume of extractant, the fraction of the aerosol sample used and the volume of air pumped for each sample, after correction for
procedural blanks. Soluble organic iodine (SOI) concentrations were calculated using Eqn. 1 (Baker, 2005). Non-seasalt ion concentrations (nss-X) and the enrichment factor of TSI with respect to seaspray ($EF_{TSI}$) were calculated using Eqns. 2 and 3 respectively (where X = $K^+$, $Ca^{2+}$, $SO_4^{2-}$ and the subscripts $A$ and $sw$ refer to the aerosol and seawater phases respectively). Sodium (Na) was used the tracer of seaspray content in the aerosol in these calculations.

$$SOI = TSI - (I^- + IO_3^-) \qquad \text{Eqn. 1}$$
$$nss\text{-}X = X_A - Na_A \, X_{sw} / Na_{sw} \qquad \text{Eqn. 2}$$
$$EF_{TSI} = (TSI / Na_A) / (I_{sw} / Na_{sw}) \qquad \text{Eqn. 3}$$

Where the magnitude of the propagated error (standard deviation) in the calculated parameter (SOI or nss-X) was greater than the magnitude of the calculated parameter itself, the calculated parameter was considered to be unreliable and was excluded from further analysis. Note that some negative values for SOI remain in the dataset once these unreliable values are
removed. Although such negative concentrations are not plausible, they have been retained in order to avoid biasing the dataset. This approach is similar to that used in the analogous determination of soluble organic nitrogen concentration, as discussed by Lesworth et al. (2010).

### 2.4 Air mass back trajectories

Air mass back trajectories for 5 day periods at heights of 10, 500 and 1000 m above the ship's position at the start, middle
and end of each sampling period were obtained from the NOAA READY Hysplit model (Stein et al., 2015).



## 3. Results and Discussion

### 3.1 Air Mass Types

The major air mass types encountered during AMT21 were similar to those reported for earlier AMT cruises (Baker et al., 2006), with air arrivals from Europe (EUR), North Africa (SAH) and southern Africa (SAF, or SAB if biomass burning tracers were present), as well as air that had passed over the North and South Atlantic (RNA and RSA, respectively) for the preceding 5 days. Examples of back trajectories for these airmass types are shown in Fig. 1.

### 3.2 Background Aerosol Composition

The major influences on aerosol composition during the AMT21 cruise are illustrated by the distributions of a number of MIs and TMs (Fig. 2). These species have sources from combustion of fossil fuels and biomass (nss-$SO_4^{2-}$, $NO_3^-$, oxalate and soluble V (s-V) (Narukawa et al., 1999; Agrawal et al., 2008; Lamarque et al., 2013)), mineral dust (nss-$Ca^{2+}$ and soluble Mn (s-Mn) (Nickovic et al., 2012)) and natural and anthropogenic organic matter emissions (oxalate (Martinelango et al., 2007)). Marine emissions of dimethyl sulphide also contribute to the nss-$SO_4^{2-}$ load, with this biogenic source probably being a more significant contributor in the less polluted air masses at the extreme south of the transect (Lin et al., 2012).

All of the species illustrated in Fig. 2 have much lower concentrations in the South Atlantic south of 12°S (sample 27 onwards) than further north. This reflects the much smaller area of the land masses in the southern hemisphere and the dominance of terrestrial sources for the species in question. Combustion sources appear to be a significant influence on airmasses originating in Europe, North Africa and Southern Africa during the cruise, as indicated by the distributions of nss-$SO_4^{2-}$, $NO_3^-$, oxalate and s-V (Fig. 2a, b, d & f) and the airmass back trajectories shown for samples 12, 17, 20 and 24 in Fig. 1. The relatively high concentrations of s-V in samples 5, 7 and 10 may indicate that the corresponding relatively high concentrations of nss-$SO_4^{2-}$ and $NO_3^-$ in these samples are due to combustion of heavy fuel oils in shipping (Becagli et al., 2012). By contrast, heavy fuel oil combustion appears to be a minor component of the combustion products encountered in the South Atlantic (Fig. 2f).

The high (relative to other southern hemisphere samples) concentrations of nss-$SO_4^{2-}$, $NO_3^-$ and oxalate and low concentrations of s-V in samples 24-26 (Fig. 2a, b, d, f) may indicate the presence of biomass burning products from southern Africa in these samples. Fine mode aerosol nss-$K^+$ concentrations in these samples (0.4 – 1.3 nmol m$^{-3}$) were at least 4-fold higher than in the other southern hemisphere samples, which would also be consistent with the influence of biomass burning (Andreae, 1983; Baker et al., 2006).



**Figure 2.** Fine (<1 μm) and coarse (>1 μm) aerosol concentrations of a) NO$_3^-$, b) nss-SO$_4^{2-}$, c) nss-Ca$^{2+}$, d) oxalate, e) s-Mn and f) s-V during the AMT21 cruise. For samples 15 and 30, the fine and coarse fractions are the sum of impactor stages 5 & 6 and the backup filter and impactor stages 1 – 4, respectively. Where concentrations were below detection limit, these are indicated by unfilled bars whose magnitude corresponds to 75% of the detection limit.



Plumes of mineral dust originating in the arid regions of the Sahara and Sahel appear to be ubiquitous in the latitude range 10 - 25°N during the months of October / November (e.g. Losno et al., 1992; Powell et al., 2015; Baker and Jickells, 2017). This was also the case during AMT21, but dust was also present further north, as indicated by relatively high concentrations of both nss-$Ca^{2+}$ and s-Mn (Figs 2c & e). In the case of samples 10-12, dust appears to be present with much higher proportions of combustion products ($NO_3^-$, nss-$SO_4^{2-}$, oxalate, s-V) than in the other dusty samples. The majority of lower

level (10 and 500 m) trajectories for these samples originate over the Iberian Peninsula, so that these samples appear to contain aerosols derived from European pollution, mixed with dust that has settled from higher altitudes.

### 3.3 Iodine Distribution and Speciation

The gradient in TSI concentrations between the northern and southern hemispheres (Fig. 3a) is much less pronounced than was observed for the predominantly terrestrial-sourced species shown in Fig. 2. This is consistent with marine emissions

being the dominant source of iodine to the atmosphere (Saiz-Lopez et al., 2012). However, TSI concentrations in the northern hemisphere (19 - 103 pmol m$^{-3}$) were higher than in the southern hemisphere (12 - 44 pmol m$^{-3}$), which is presumably a consequence of the greater flux of iodine containing gases from the sea surface, driven by higher northern hemisphere MBL ozone concentrations (Prados-Roman et al., 2015; Gomez Martin et al., 2020). Concentrations of TSI, iodine species and EF$_{TSI}$ observed during AMT21 were all similar to those observed during previous cruises in the Atlantic

(Table 2).

Iodate is the dominant form of soluble iodine in the coarse mode in all airmass types and is also a substantial component of the fine mode iodine in the RNA, SAH and RSA types (Fig. 4b). For all airmass types both I$^-$ and SOI have higher proportions in the fine mode than the coarse mode (Fig. 4a & c). It appears that in the airmass types with low fine mode iodate (EUR, SAF & SAB), the proportions of both the fine mode I$^-$ and SOI forms are greater than in the other airmass

types. It is notable that the EUR, SAF and SAB types also have lower proportions of $IO_3^-$ in their coarse fractions than the other airmass types. These patterns suggest that there may be some systematic differences in aerosol chemistry that influence iodine speciation, but it should be noted that, with the exception of the SAH and RSA types, relatively few (2 – 4) samples of each type were collected during AMT21. Some caution may be necessary when interpreting the iodine speciation in these poorly-sampled airmass types.





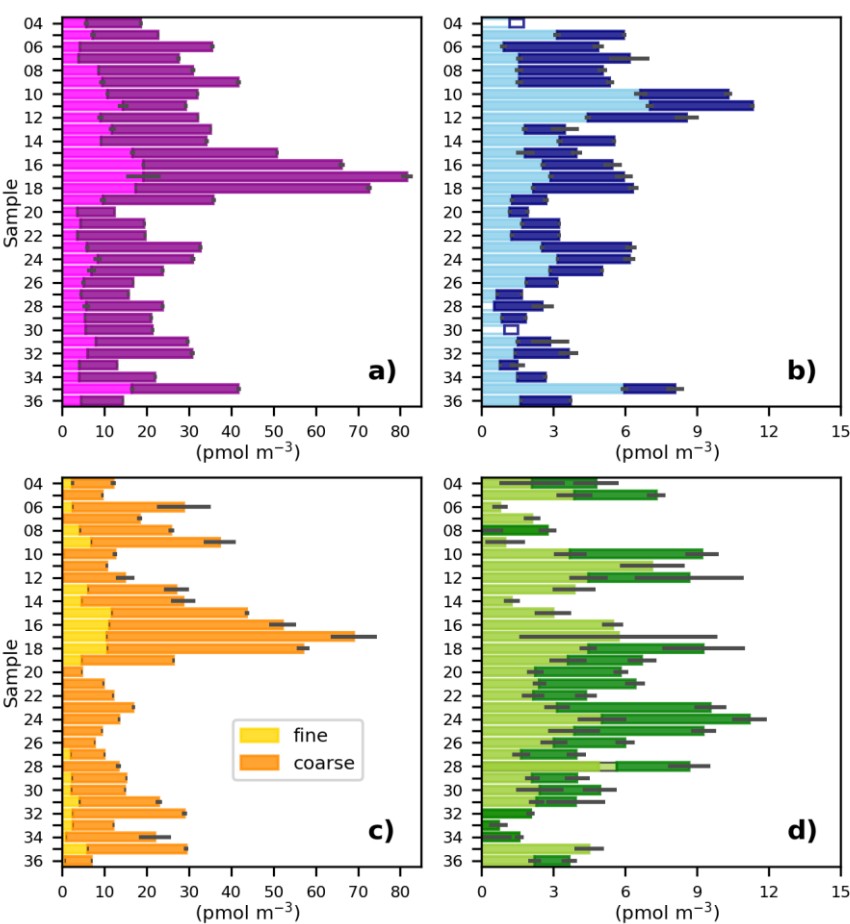


**Figure 3.** Fine (<1 μm) and coarse (>1 μm) aerosol concentrations of a) TSI, b) I⁻, c) IO₃⁻, and d) SOI during the AMT21 cruise. Details are as for Figure 2.





**Table 2. Summary (median and range) of fine (< 1 µm) and coarse (> 1 µm) mode aerosol iodine concentrations (pmol m⁻³) and enrichment factors for TSI (EF$_{TSI}$) observed during AMT21 and other cruises in the Atlantic Ocean (M55, AMT13 (Baker, 2005), RHaMBLe (Allan et al., 2009)).**

| Cruise | AMT21 | | M55 | | AMT13 | | RHaMBLe | |
|---|---|---|---|---|---|---|---|---|
| Species | < 1 µm | > 1 µm | < 1 µm | > 1 µm | < 1 µm | > 1 µm | < 1 µm | > 1 µm |
| TSI | 6.9 | 21.7 | 7.5 | 11.4 | 8.4 | 13.4 | 13.4 | 23.2 |
| | (3.6 – 19.2) | (8.8 – 62.4) | (4.4 – 49.4) | (3.2 – 56.6) | (3.3 - 19.6) | (5.8 – 44.9) | (9.1 – 25.5) | (7.4 – 61.4) |
| Iodide | 1.65 | 2.15 | 0.92 | 1.65 | 3.34 | 3.89 | 3.30 | 2.87 |
| | (<0.53 – 7.03) | (<0.55 – 4.60) | (<0.39 – 3.62) | (0.74 – 6.85) | (0.99 - 13.5) | (0.61 – 11.3) | (0.88 – 11.4) | (<0.11 – 8.71) |
| Iodate | 2.23 | 14.67 | 0.98 | 2.92 | 0.89 | 7.59 | 2.04 | 18.99 |
| | (<0.17 – 11.8) | (4.44 – 58.4) | (0.39 – 15.7) | (<0.86 – 47.0) | (0.44 - 4.73) | (<0.65 – 42.4) | (<1.27 – 34.8) | (4.36 – 47.4) |
| SOI | 2.76 | 2.97 | 6.04 | 5.41 | 3.80 | 1.64 | 8.75 | 3.39 |
| | (0.72 – 7.14) | (-4.12 – 6.41) | (3.04 – 30.2) | (1.49 – 13.6) | (1.60 - 8.94) | (-4.64 – 5.69) | (-26.6 - 13.0) | (-31.9 – 9.91) |
| EF$_{TSI}$ | 648 | 196 | 2144 | 158 | 2402 | 94 | 1833 | 104 |
| | (244 - 1540) | (42 - 1090) | (831 - 12800) | (47 - 1230) | (693 - 13100) | (28 - 326) | (914 - 5320) | (49 - 326) |

Pechtl et al. (2007) suggested that aerosol IO$_3^-$ may be reduced under acidic conditions. Although it has recently become

possible to measure aerosol pH directly (Craig et al., 2018) or to calculate this parameter using speciation modelling with supporting aerosol and gas phase composition measurements (e.g. Pye et al., 2020), estimates of aerosol pH are not available for AMT21. Nevertheless, the observed variations in the proportion of IO$_3^-$ between aerosol size fractions and airmass types is consistent with the results of the Pechtl et al. modelling study. Iodate proportions were highest in the coarse modes of the RNA, SAH and RSA types, in which alkaline conditions are expected due to the presence of fresh sea spray or mineral dust

aerosols. Fine mode aerosols are generally acidic (Pye et al., 2020) and fine mode IO$_3^-$ proportions were very low in airmass types that contained high concentrations of acidic pollutants (e.g. nss-SO$_4^{2-}$, Fig. 4d) from Europe and southern Africa (EUR, SAF & SAB). Relatively high concentrations of acidic species were also present in the fine mode of SAH-type aerosols (Fig. 4d), but this aerosol fraction also contained alkaline mineral dust (Fig. 2c) was likely to be less acidic than the EUR, SAF & SAB fine fractions. Reduction of IO$_3^-$ produces HOI, which has the potential to react with organic matter, forming SOI and

eventually I$^-$ (Baker, 2005; Pechtl et al., 2007). The distribution of SOI and I$^-$ in fine mode aerosols and the high relative





abundance of these species in the EUR, SAF and SAB fine modes are therefore consistent with the combined mechanism of $IO_3^-$ reduction under acidic conditions and subsequent generation of SOI and $I^-$ proposed by Pechtl et al. (2007).

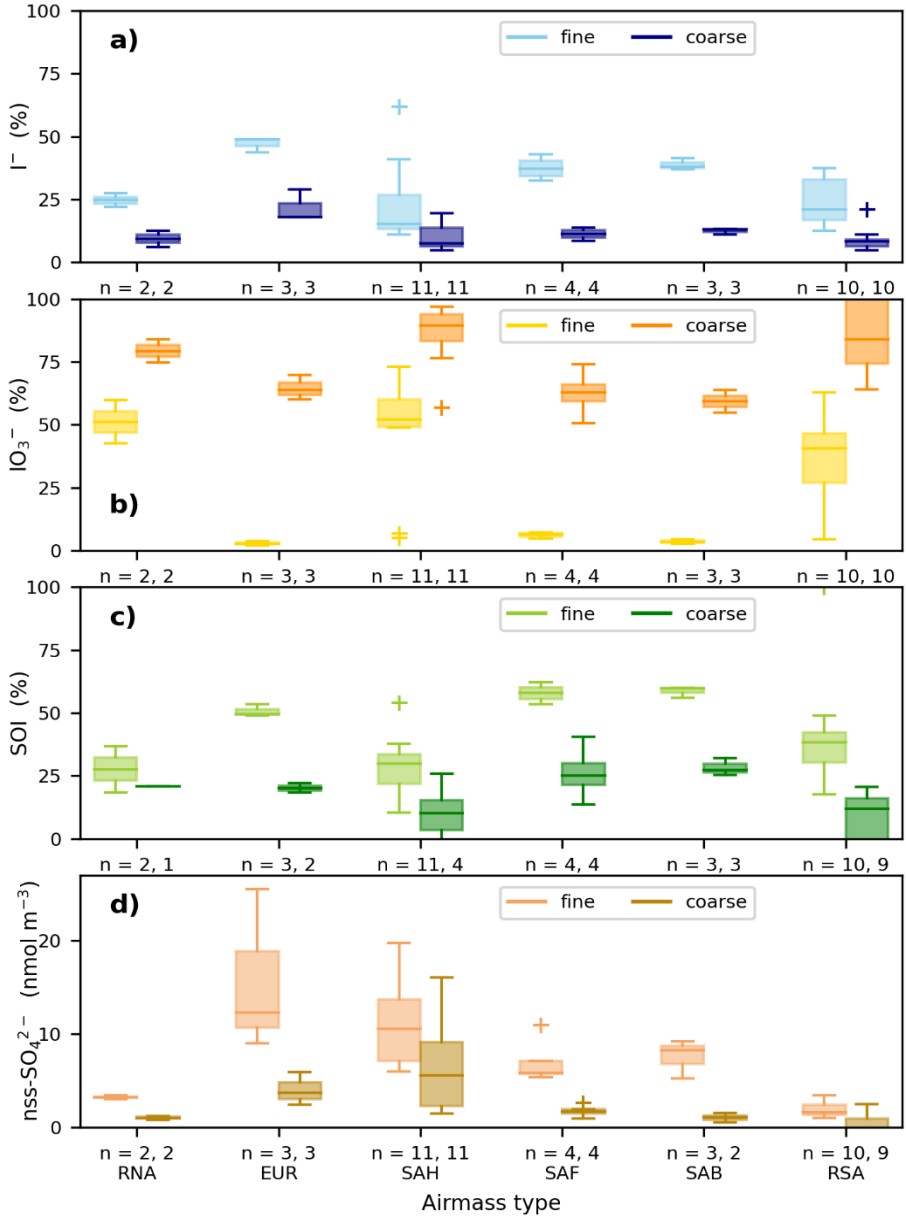

**Figure 4.** Proportions of a) $I^-$, b) $IO_3^-$, and c) SOI (expressed as percentage of TSI) and d) the concentrations of nss-$SO_4^{2-}$ in fine (<1 μm) and coarse (>1 μm) aerosols in the different air mass types encountered during AMT21. The number of samples in fine and coarse fractions respectively for each group (n) is also shown. In some cases, unreliable results for SOI and nss-$SO_4^{2-}$ in the coarse mode were excluded, as described in the text. Boxes show the inter-quartile range of the data, with horizontal lines representing the median. Whiskers show the range of the data, except where outliers more than 1.5 times the inter-quartile range beyond each quartile are present (crosses).






Oxalate has been reported to be an end product of the photochemical oxidation of organic matter in aerosols (Kawamura and Ikushima, 1993). Although this species is unlikely to be iodinated, it is used here to infer the presence of organic matter that can be iodinated in the AMT21 samples. There are statistically significant relationships between oxalate and SOI in both the fine and coarse modes ($r^2 = 0.36$ and 0.47 respectively, both $p < 0.01$), with SOI : oxalate ratios of ~ $3 \times 10^{-3}$ mol mol$^{-1}$. Low
molecular weight organo-iodine compounds (e.g. iodoacetic acid, diiodoacetic acid , iodopropenoic acid) that have been reported in aerosols (Yu et al., 2019) may form part of the SOI fraction determined here.

### 3.4 Iodine Speciation in Mineral Dust Aerosols

The highest concentrations of TSI encountered during AMT21 were in the samples collected at 11 - 23 °N (samples 15 – 19), which contained high concentrations of mineral dust, with the majority of the iodine contained in these samples being in the
form of coarse-mode $IO_3^-$ (Fig. 3c). Figure 5a shows the relationship between the total (fine plus coarse) concentrations of $IO_3^-$ and nss-$Ca^{2+}$ in the AMT21 samples. There was a significant correlation between these parameters for samples of Saharan origin (SAH) during the cruise. Relative enrichment of coarse-mode $IO_3^-$ in aerosol samples containing Saharan dust has been noted in several previous studies (Baker, 2004, 2005; Allan et al., 2009), but the $IO_3^-$ : nss-$Ca^{2+}$ ratio was more variable during these earlier cruises in the Atlantic (Fig. 5b). There are a number of potential explanations for the occurrence
of high concentrations of $IO_3^-$ in mineral dust aerosols.

Iodate may be contained in the mineral dust at the point of uplift from its parent soils, or it may accumulate on the dust aerosol during atmospheric transport, presumably by condensation from the gas phase. As noted above, the alkalinity (specifically the carbonate content, as indicated by nss-$Ca^{2+}$) of mineral dust inhibits the reduction of $IO_3^-$ (Pechtl et al., 2007), so that this species is expected to be stable and accumulate in mineral dust aerosol. Alkalinity may also play a role in
promoting the uptake of acidic species, such as iodic acid ($HIO_3$; Plane et al., 2006) from the gas phase.

Figure 6 shows the distributions of oxalate, nss-$Ca^{2+}$ and the iodine species in the multi-stage impactor sample collected in the North Atlantic (sample 15). As was the case for the AMT21 samples in general (Fig. 3), $IO_3^-$ was the dominant iodine species in most size fractions (Fig. 6d). Iodide was not detectable in several fractions (Fig. 6c) and reliable SOI concentrations could not be determined in 4 of the 7 fractions (Fig. 5e). TSI was enriched, relative to seawater
concentrations, in all size fractions ($EF_{TSI} > 200$), but was most strongly enriched in stages 4 & 5 (aerodynamic diameters 0.61 – 1.6 µm).





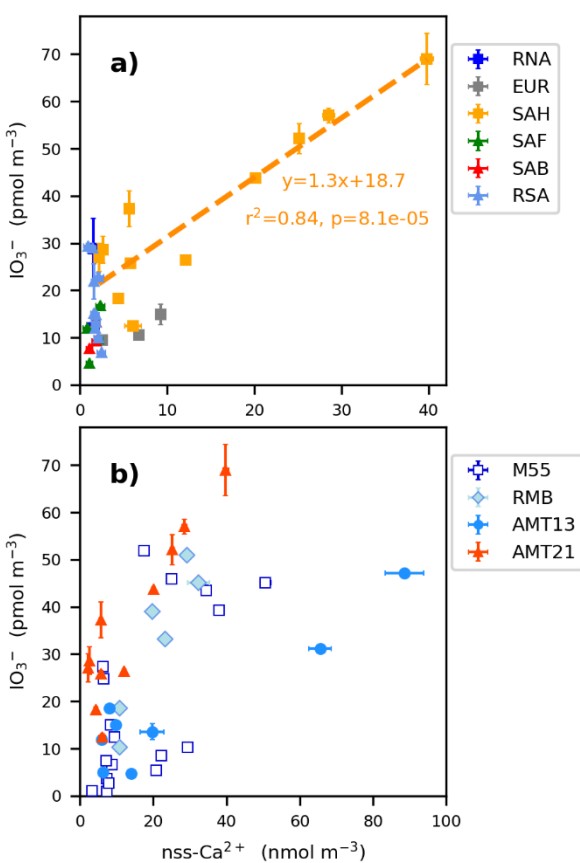

**Figure 5. Total (sum of fine and coarse) aerosol IO$_3^-$ concentrations as a function of nss-Ca$^{2+}$ concentrations for a) all of the airmass types encountered during AMT21 and b) for SAH-type samples during the M55, RHaMBLe (RMB), AMT13 and AMT21**

**cruises. Airmass type codes are described in the text.**

The size distribution of IO$_3^-$ in sample 15 (Fig. 6d) is dissimilar to that of the dust tracer nss-Ca$^{2+}$ (Fig. 6b) and to the size distributions of other elements associated with mineral dust (Fe, Al, Mn, Ti, Co, Th) in a size fractionated aerosol sample collected concurrently with sample 15 (Baker et al., 2020). All of these dust tracers showed maximum concentrations in the

larger size fractions (Stages 1 & 2) than observed for IO$_3^-$ (maximum Stage 4). In the absence of information on the iodine content of desert dust source materials, the potential contribution of dust to the observed aerosol IO$_3^-$ concentrations has been estimated using the ratio of I : Al in shale (5.85 x 10$^{-6}$ mol mol$^{-1}$; Turekian and Wedepohl, 1961). This suggests that dust contributes < 2% of the observed TSI concentration in Stages 1 – 5 of sample 15, using the total Al concentrations reported for these fractions by Baker et al. (2020). These differences in particle size distribution and the low direct potential

contribution to observed IO$_3^-$ concentrations from mineral dust itself suggest that dust is not the principal source of I in this case.



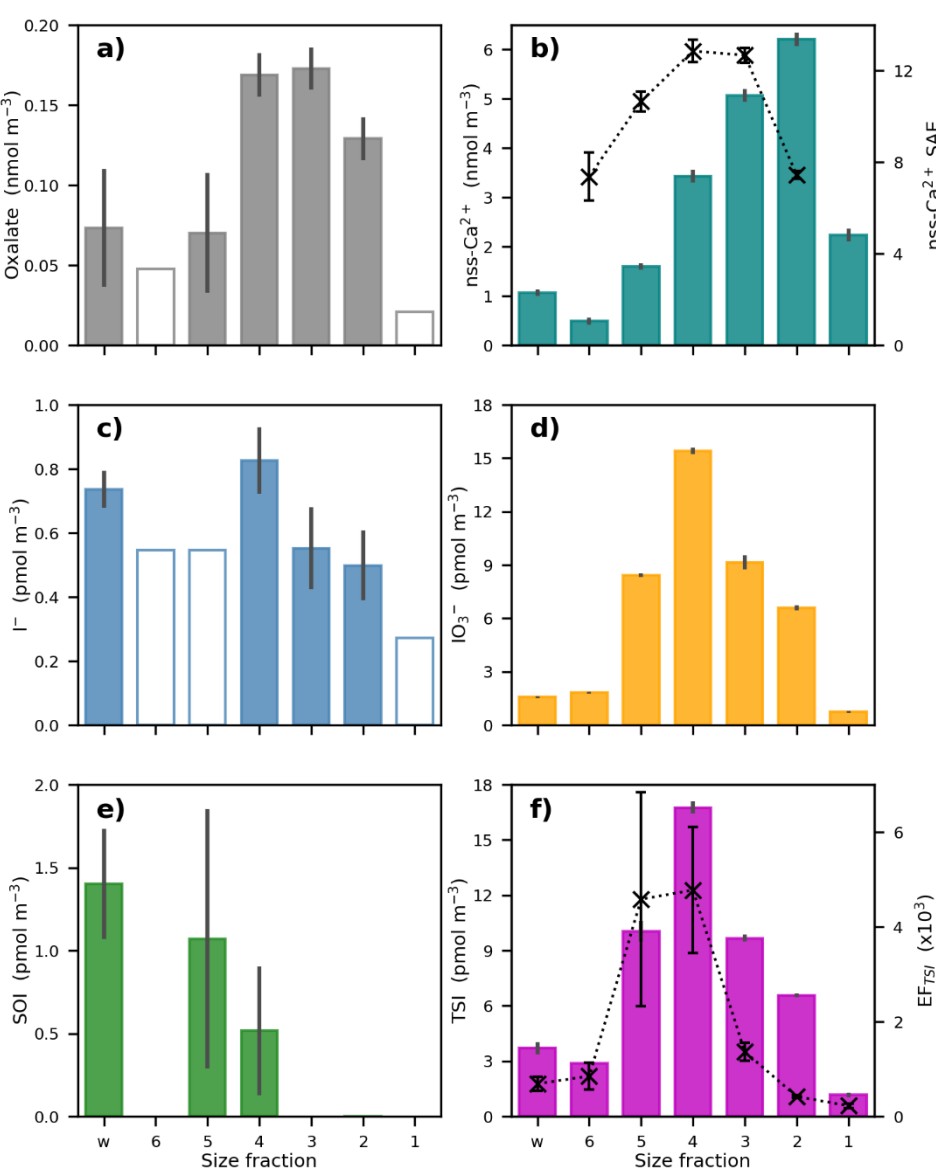

**Figure 6. Concentrations of a) oxalate, b) nss-Ca$^{2+}$, c) I$^-$, d) IO$_3^-$, e) SOI and f) TSI in the size fractions of sample 15. (w indicates the backup filter and particle size increases from stage 6 to stage 1 – stage cut-off diameters are given in the text). Crosses in b) show the surface area equivalent (SAE) of nss-Ca$^{2+}$ (arbitrary units) for impactor stages 2-6 and in f) show the enrichment factors of TSI in each stage.**

Figure 6b also shows a rough estimate of the surface area distribution of CaCO$_3$ aerosol in Stages 2 – 6 of sample 15. This

parameter (nss-Ca$^{2+}$ SAE) was calculated assuming that surface area is proportional to the ratio 3 / r, where r is particle





radius. In this calculation, r was taken from the modal particle size for each impactor stage under the flow rate used for sampling (2.5, 1.2, 0.8, 0.45, 0.2 μm for Stages 2 – 5 respectively). This relationship (i.e. the assumption of spherical geometry) is not expected to be realistic, but it is to be expected that the distribution of $CaCO_3$ surface area will have a maximum at smaller particle sizes than the distribution of $CaCO_3$ mass. The observed distribution of $IO_3^-$ concentrations (Fig. 6d) may therefore be more consistent with uptake (of $HIO_3$) onto alkaline dust surfaces than with $IO_3^-$ being a

constituent of uplifted dust.

## 4. Conclusions

    The results presented above appear to be in broad agreement with the behaviour predicted by Pechtl et al. (2007) for sulphate and seasalt aerosols in which the speciation of iodine is controlled by a combination of acid-dependent reduction of $IO_3^-$ and the production of $I^-$ via the reaction of HOI with dissolved organic matter. The alkalinity associated with mineral dust may

also contribute to the high concentrations of $IO_3^-$ found in Saharan dust aerosols in this and other studies (Baker, 2004, 2005; Allan et al., 2009). However, the AMT21 dataset is insufficient to unambiguously confirm the role of acidity or the HOI – organic matter reaction in controlling iodine speciation in aerosols. In particular, it was not possible to determine aerosol pH during the cruise, so the relationship between this parameter and the observed iodine speciation remains unclear. Pechtl et al. (2007) noted that detailed laboratory studies on the reactions between HOI and organic matter under conditions relevant to

aerosols were required and this is still the case. Such studies, together with detailed, simultaneous field observations of iodine speciation in the aerosol- and gas-phases and aerosol pH will be required in order to make further progress towards understanding the controls on atmospheric iodine speciation and its impact on ozone chemistry.

    Prados-Roman et al. (2015) suggest that the enhanced emission of volatile iodine from the sea surface caused by increasing pollutant ozone since the pre-industrial era represents a negative feedback on atmospheric ozone, because higher

atmospheric iodine concentrations enhance the rate of ozone destruction. Atmospheric acidity has also changed significantly over the industrial era. Aerosol pH has declined (by up to 2 pH units over the mid-latitude North Atlantic) due to anthropogenic emissions of acidic pollutants and is expected to increase in the future in response to changes in those emissions (Baker et al., 2021). Whether those changes in pH are substantial enough to alter the recycling of I to the gas phase, and hence to change ozone destruction rates over the ocean, may also merit further investigation.

## Data availability


    The data used in this work is available from the British Oceanographic Data Centre.



**Author Contributions**

The study was conceived by ARB. Sampling was conducted by CY, who also determined major ion and iodine concentrations. ARB analysed samples for trace metal composition. Both authors contributed to data interpretation and the
manuscript was drafted by ARB with contributions from CY.

**Competing interests**

The authors declare that they have no conflict of interest.

**Acknowledgments**

CY is indebted to the Royal Thai Government for provision of a Research Studentship. Instrumental analysis was partly
supported by the UK GEOTRACES programme, through grant number NE/H00548X/1 of the UK Natural Environment Research Council. The authors gratefully acknowledge the NOAA Air Resources Laboratory (ARL) for the provision of the READY website (https://www.ready.noaa.gov) used in this publication.

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
