# Peer review of "Measurement report: Indirect evidence for the controlling influence of acidity on the speciation of iodine in Atlantic aerosols"

_Atmospheric Chemistry and Physics, 2021_

## Author Comment (AC1)

This paper presents an investigation of the speciation of water-soluble iodine in Atlantic aerosol collected during a cruise in Sept-Nov. 2011. The results and relative discussion is very interesting for a better understanding of formation and conversion of iodine species in aerosol. The paper was well presented and fit to the scope of the journal well, it is there recommended for publication after some revision.

We thank the reviewer for their thoughtful consideration of our work.

1. Only water-soluble iodine in aerosol was investigated in this work, not total iodine in the aerosol. This issue might need to be clarified in the title or abstract.

As suggested, we have altered the Abstract. The first sentence now reads "The speciation of soluble iodine and ...".

2. It was well presented that marine emission is the dominant source of iodine in the atmosphere and the higher TSI Atlantic aerosol in the North Hemisphere was attributed to the higher MBL ozone concentrations in the North Hemisphere compared to the South Hemisphere. Meanwhile, the high iodate in the samples No. 15-19 was explained as the consequence of high concentrations of mineral dust originated from Sahara though uptaking HIO3 from air. So, where does the HIO3 in the air come from? How was marine emitted iodine converted to HIO3, even in acidic condition?

There appears to be a sequence of reactions in the gas phase that lead to the formation of HIO3:

$I + O_3 \leftrightarrow IO + O_2$

$IO + IO \leftrightarrow OIO + I$          (Cox et al., 1999)

$OIO + OH \leftrightarrow HIO_3$          (Plane et al., 2006)

These reactions are not influenced directly by aerosol acidity. We have added an explanation of how HIO3 is formed at lines 232-233.

3. One interesting point of this paper is to interpret the iodide and SOI as a result of iodate reduction in acidic condition, formation of HOI and reaction of HOI with organic substance. But, the main evidence (modeling) seems not strong enough to support this interpretation. Why iodide and SOI in the aerosol has to be produced through reduction of iodate, not directly formed during the release or formation of the aerosol? The reaction product of iodine with ozone can be reductive form of iodine (HI, I2), which can easily form SOI and iodide and associated to aerosol. Meanwhile, no strong (direct) evidence show the acidic condition in the fine particles. The different species of iodine in fine particles compared to coarse particle might be

attributed to the different of source or formation pathway of iodine species, why was the direct sea-spray source of iodate in coarse particles excluded ?

Here the reviewer raises some interesting questions about the sources and interrelations of the various iodine species discussed in the manuscript. For the most part however, we are not able to answer these questions with the data available to us. Far more detailed information on the gas phase speciation of iodine would be required for this, along with knowledge (currently lacking) of the identity and reactivity of the compounds that comprise the SOI pool. We made all of these points in the final section of the manuscript.

It has been apparent for some time that models that only take account of the inorganic chemistry of iodine (e.g. HI and I2, referred to by the reviewer) are not able to reproduce the observed speciation of iodine in MBL aerosols (see Vogt et al., 1999; McFiggans et al., 2000; Pechtl et al., 2007). We have amended the text on line 53 to emphasise this point. This now reads "Models that do not include the reduction of iodate and organic matter – iodine reactions have been unable to reproduce the observed inorganic and organic speciation of iodine in marine aerosols".

The reviewer is correct that SOI may not be formed in the aerosol phase. Potential sources of SOI were discussed in detail by Baker (2005). Factors considered included:

- The volatility of the SOI collected as aerosol samples,
- The size distribution of SOI (and the implications of this for its source),
- The potential for SOI to have originated from the sea surface and the available (very limited) information on enrichment of organic iodine in the sea surface microlayer.

Rather than re-visit these discussions (which have not changed), we refer the interested reader to the work of Baker (2005):

"It should be noted that the sources of SOI in marine aerosol are not yet clear and that ejection of SOI from the sea-surface microlayer during bubble bursting may contribute to the SOI observed in the AMT21 aerosol samples. See Baker (2005) for a discussion of the potential sources of aerosol SOI and the likely impacts of these sources on iodine speciation." (lines 201-204).

The reviewer is also correct that we do not have direct evidence of the acidic condition of the fine particles. We have stated this fact in the manuscript and attempted to emphasise that the interpretation of the importance of acidity in controlling iodine speciation is only indirect in the title of the paper and throughout the discussion and conclusions.

The reviewer asks how sea-spray can be excluded as a source of coarse mode iodate. The enrichment factors for total soluble iodine (EF(TSI)) presented in Table 1 are clear evidence that the vast majority of iodine in coarse aerosols is not sourced from sea-spray. Since iodate comprises the major fraction of coarse mode iodine, it is apparent that iodate must also be considerably enriched in this fraction. We did not include calculations of the enrichment factor of iodate (EF(IO3)) in the manuscript, because the proportion of dissolved iodine in surface seawater present as iodate varies strongly with latitude over the AMT21

cruise track (Chance et al., 2014; Chance et al., 2019). We are therefore not able to derive values for EF(IO3) with confidence. However, if we calculate EF(IO3) for the coarse mode assuming 100% of sea-spray iodine exists as iodate (i.e. the minimum values for EF(IO3)), the results obtained are in the range 39 – 820 (median 130). These values clearly indicate that sea-spray is a minor source of iodate to the coarse mode aerosols we collected.

4. A much lower EF(TSI) in fine particles in AMT21 (648) compared to the previous cruises (1833-2402) was presented, a discussion on it is expected.

Thank you for pointing this out. We are not able identify the cause of this difference in EF values, but we have modified the text on lines 167-170 to make sure that the reader is aware.

5. The "Conclusion" section seems not a conclusion deduced from the discussion, but a further discussion/ perspectives. It might be better to change the title of this section or change the content of this section.

We have changed the section heading to "Conclusions and Further Considerations", as suggested.

References

Baker, A. R.: Marine aerosol iodine chemistry: The importance of soluble organic iodine, Environmental Chemistry, 2, 295-298, 2005.

Chance, R. J., Baker, A. R., Carpenter, L. J., and Jickells, T. D.: The distribution of iodide at the sea surface, Environmental Science: Processes & Impacts, 16, 1841-1859, 10.1039/C4EM00139G, 2014.

Chance, R. J., Tinel, L., Sherwen, T., Baker, A. R., Bell, T., Brindle, J., Campos, M. L. A. M., Croot, P., Ducklow, H., Peng, H., Hopkins, F., Hoogakker, B., Hughes, C., Jickells, T. D., Loades, D., Macaya, D. A. R., Mahajan, A. S., Malin, G., Phillips, D., Roberts, I., Roy, R., Sarkar, A., Sinha, A. K., Song, X., Winkelbauer, H., Wuttig, K., Yang, M., Peng, Z., and Carpenter, L. J.: Global sea-surface iodide observations, 1967–2018, Scientific Data, 6, 286, 10.1038/s41597-019-0288-y, 2019.

Cox, R. A., Bloss, W. J., Jones, R. L., and Rowley, D. M.: OIO and the atmospheric cycle of iodine, Geophysical Research Letters, 26, 1857-1860, 1999.

McFiggans, G., Plane, J. M. C., Allan, B. J., Carpenter, L. J., Coe, H., and O'Dowd, C.: A modeling study of iodine chemistry in the marine boundary layer, Journal of Geophysical Research, 105, 14371-14385, 2000.

Pechtl, S., Schmitz, G., and von Glasow, R.: Modeling iodide - iodate speciation in atmospheric aerosol: Contributions of inorganic and organic iodine chemistry, Atmospheric Chemistry and Physics, 7, 1381-1393, 2007.

Plane, J. M. C., Joseph, D. M., Allan, B. J., Ashworth, S. H., and Francisco, J. S.: An experimental and theoretical study of the reactions OIO plus NO and OH plus OH, Journal of Physical Chemistry A, 110, 93-100, 2006.

Vogt, R., Sander, R., von Glasow, R., and Crutzen, P. J.: Iodine chemistry and its role in halogen activation and ozone loss in the marine boundary layer: A model study, Journal of Atmospheric Chemistry, 32, 375-395, 1999.

---

## Author Comment (AC2)

The authors report the results of quite detailed analyses of aerosol composition collected during a ship expedition on the Atlantic between England and the southern tip of South America. The main analyte is iodine, in particular the speciation of particle-bound iodine into four fractions (iodide, iodate, total iodine, soluble organic iodine). For the interpretation of the results a number of other components (e.g. trace metals) are also determined. The results of the sophisticated analysis are also evaluated with the use of back trajectories. Since different size fractions are always sampled, there is also particle size-relevant information, which in turn is used by the authors to interpret sources. One of the main conclusions is the influence of aerosol acidity on the chemistry and thus the occurrence of the different iodine species in atmospheric aerosol particles. The manuscript is well written, is based on an excellent data set, and the conclusions drawn are well justified. The topic is certainly relevant to ACP readers and therefore I recommend publication with only minor changes.

We thank the reviewer for their careful consideration of our manuscript and helpful comments.

Page 5, line 94: missing word !?

We have rewritten the text here. The amended text is:

"For the determination of soluble TMs, samples were extracted into ~ 1 M ammonium acetate solution followed by analysis using inductively coupled plasma – optical emission spectroscopy (ICP-OES). Full details of these methods have been reported in Baker and Jickells (2017)."

Page 5, line 103: an "as" is missing

This has been inserted.

The authors discuss in several passages the iodine chemistry, e.g. the reactions of IO3- to HOI, SOI and I- and quote from Baker 2005 and Pechtl et al. 2007. Since without any question the speciation is in the center of the work and also in the title the possible influence of the chemistry on the iodine species is pointed out, I would find it helpful for the reader to compile the central chemical reactions from Baker 2005 and Pechtl et al. 2007 for this paper. In doing so the understanding of the discussion would be simplified from my point of view.

Thank you for this suggestion. The most important reactions in this context are the acid-dependent reduction of iodate:

$IO_3^- + I^- + 2H^+ \leftrightarrow HIO_2 + HOI$

And the formation of SOI and its subsequent decay to iodide:

$HOI + DOM \leftrightarrow SOI \leftrightarrow DOM + I^- + H^+$

The latter reaction was included as $HOI + DOM \leftrightarrow DOM + I^- + H^+$ by Pechtl et al. because they did not include an organic iodine fraction (SOI) in their model. We have added these reactions at lines 56 – 58. We have also amended the description of the Pechtl et al. model on lines 51 – 53, as the original description of this was slightly inaccurate.